# Dynamic machine vision with retinomorphic photomemristor-reservoir computing

Hongwei Tan [1] ✉ & Sebastiaan van Dijken [1] ✉

Dynamic machine vision requires recognizing the past and predicting the future of a moving object based on present vision. Current machine vision systems accomplish this by processing numerous image frames or using complex algorithms. Here, we report motion recognition and prediction in recurrent photomemristor networks. In our system, a retinomorphic photomemristor array, working as dynamic vision reservoir, embeds past motion frames as hidden states into the present frame through inherent dynamic memory. The informative present frame facilitates accurate recognition of past and prediction of future motions with machine learning algorithms. This in-sensor motion processing capability eliminates redundant data flows and promotes real-time perception of moving objects for dynamic machine vision.

Dynamic machine vision (DMV) technology has numerous significant applications in video analysis, robotic vision, self-driving technology, and intelligent transport[1,2]. The ability to use present vision to recognize past motion and predict future trajectories is crucial in DMV[3,4]. Current imaging systems utilize multiple modules, including sensors, signal converters, memory, and processors, to recognize and predict motion by analyzing massive frame-by-frame image sequences and using complex algorithms[5,6], engendering redundant data flows and high-energy consumption.

Different from modern image sensing and processing systems, the biological architecture of human vision is highly capable of recognizing and predicting motion, for instance, aiding humans in the perception of danger in wildlife or traffic[7–9]. In recent years, inspired by the biological vision system wherein visual short-term memory plays a key role[10,11], retinomorphic image sensors with memory capability[1], such as switchable photovoltaic sensors[12], non-volatile phototransistors[13,14], and memristors[15], have shown adaptive and all-in-one sensing capability, facilitating in-sensor computing, self-adaptive imaging, and motion detection. Besides, in-sensor reservoir computing systems with spatiotemporal processing capabilities have been demonstrated for language learning[15] and image classification[16]. However, motion recognition and prediction (MRP) within a compact dynamic sensing system, which is crucial for DMV technology, has not been realized yet.

Here, we report recurrent photomemristor networks consisting of a retinomorphic photomemristor array (PMA) operating as a dynamic vision reservoir and readout networks for processing (Fig. 1a). In the retinomorphic photomemristor-reservoir computing (RP-RC) system (Fig. 1b), the inherent dynamic memory of the PMA stores spatiotemporal information of a frame-by-frame visual sequence as hidden states ($h$) in the last frame. The dynamic PMA reservoir, containing all the past spatiotemporal visual information, is used for various dynamic processing tasks through the training of readout networks. To demonstrate the spatiotemporal processing capability of the RP-RC system, we implement the classification of videos playing English words ending with the same letter but with different spatiotemporal dynamics for language learning. Furthermore, we realize the most crucial DMV task−motion recognition and trajectory prediction−in the RP-RC system using classification and inherent memory association by the readout networks, providing a promising neuromorphic platform for in-sensor DMV.

## Results

### Retinomorphic photomemristor-reservoir computing (RP-RC) system

Hardware-based DMV requires retinomorphic sensors with inherent dynamic processing and memory[1]. Photomemristors that integrate sensing, processing, and memory capabilities[17,18] are an ideal candidate for this task. In recent years, photomemristors have been studied in neuromorphic vision and processing systems for image classification[19–23] and human action recognition[23]. Here, we exploit photomemristors for MRP based on an informative frame with embedded memorized information from multiple previous frames.

[1]NanoSpin, Department of Applied Physics, Aalto University School of Science, P.O. Box 15100, FI-00076 Aalto, Finland. ✉e-mail: hongwei.tan@aalto.fi; sebastiaan.van.dijken@aalto.fi

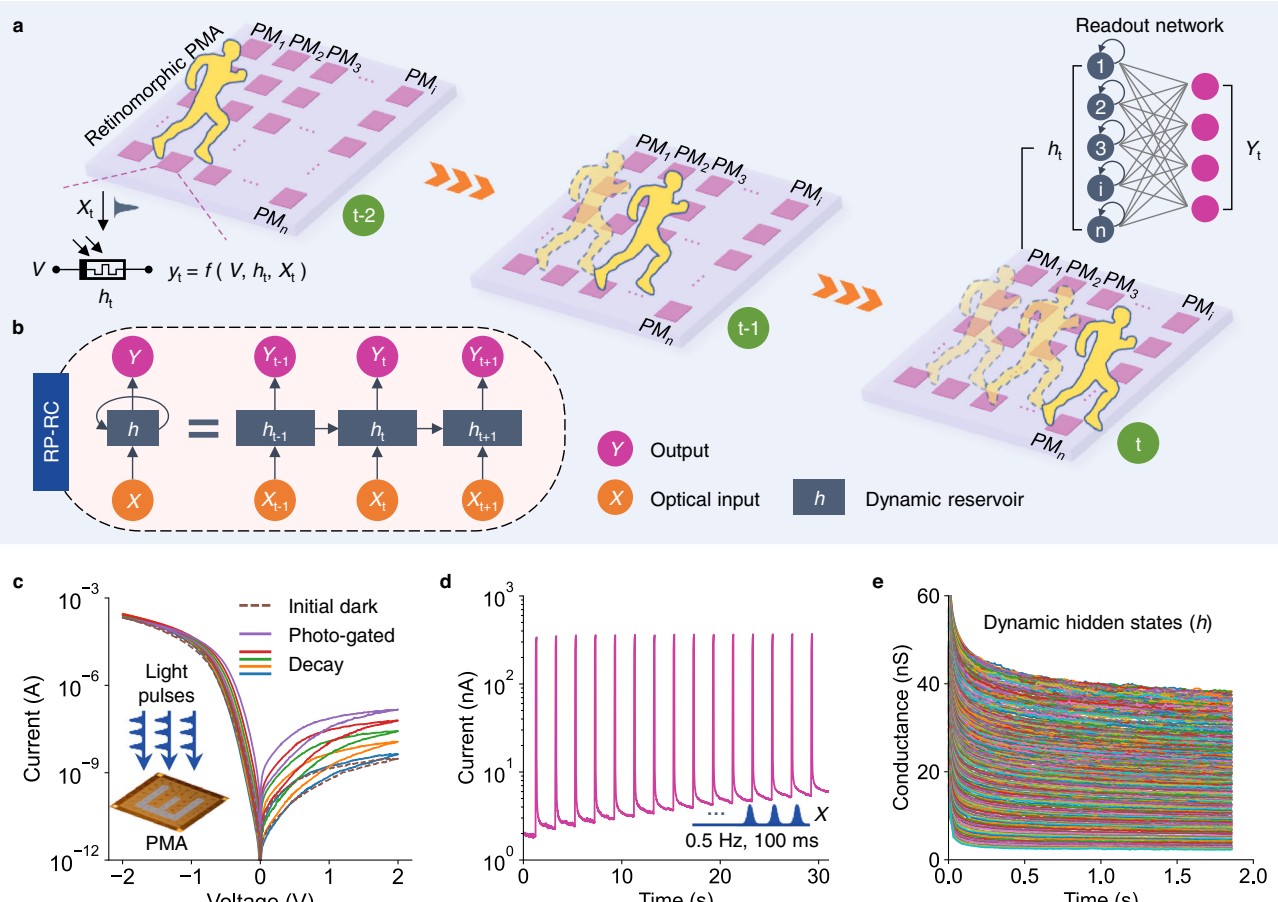

**Fig. 1 | Retinomorphic photomemristor-reservoir computing (RP-RC) system.**
**a** Schematic of the RP-RC system with a retinomorphic photomemristor array (PMA) operating as dynamic vision reservoir and a readout network. The PMA senses multiple frames of a video and temporally stores them in the last frame for further processing in the readout network. **b** Structure of the RP-RC system. $X$, $h$, and $Y$ indicate the optical inputs, hidden states, and output currents.
**c** Photomemristive switching behavior of the PMA. The illumination time is 20 s and the decay curves are measured about 1, 3, 7, 20 min after the light source is turned off. The inset illustrates the input of programmed light pulses and an optical image of the PMA. **d** Sensing, accumulating, and memory of optical inputs ($X$) using 100 ms pulses at a repetition rate of 0.5 Hz and a photomemristor bias voltage of 1.0 V. The PMA senses and temporally memorizes optical information through the slowly decaying photocurrent. **e** 300 dynamic analog hidden states of the retinomorphic PMA measured after applying 1 to 300 optical pulses with a duration of 100 ms at a repetition rate of 0.5 Hz.

To implement the RP-RC system, we fabricated a 5 × 5 PMA (inset of Fig. 1c) with an indium tin oxide (ITO)/ZnO/Nb-doped $SrTiO_3$ (NSTO) structure. In the photomemristors, optically and electrically controlled charging and migration of oxygen vacancies changes the Schottky barrier at the ZnO/NSTO interface[20], triggering a dynamic optoelectronic memristive response (Supplementary Fig. 1). As shown in Fig. 1c–e, illumination of the photomemristors by light increases the output current by 2–3 orders of magnitude and the signal decays gradually after the light is switched off (Fig. 1c). The dynamic states and decay time depend on the number of optical pulses (Supplementary Fig. 2). The high on/off ratio of ~$10^2$ in response to 100 ms light pulses (Supplementary Fig. 1b and Supplementary Table 1) and the wide continuous range of dynamic analog states (Fig. 1d, e and Supplementary Fig. 2) attained by light stimulation enables photosensing with inherent dynamic memory as rich hidden states, which is an essential requirement for retinomorphic sensors in MRP. To test the speed of photosensing with memory by the PMA, we increased the frequency of the optical input to 60 Hz (Supplementary Fig. 3), corresponding to the frequency of commercial displays. Again, a large continuous range of dynamic analog states is measured up to hundreds of optical pulses, demonstrating adequate in-sensor memory for the processing of hidden states. Finally, we note that the photomemristive response of

the PMA is highly uniform (Supplementary Figs. 4 and 5). While readout training could in part compensate for device-to-device variations in the output current, this hardware feature is relevant for complex DMV tasks.

## Spatiotemporal processing of the RP-RC system
To demonstrate dynamic vision recognition by our RP-RC system, we used videos playing words letter-by-letter as input and a simple readout network to classify the spatiotemporal information (Fig. 2a). The input words in the videos are 'APPLE', 'LIME', 'OLIVE', 'DATE', and 'GRAPE', which all end with the letter 'E'. In conventional image sensors, the last frame of the videos (the letter 'E') would be similar for all words (Supplementary Fig. 6). However, in our retinomorphic PMA, the last frame ($h_5$) does not only contain information on the last letter 'E', but also of all previously played letters because of its inherent dynamic memory (Supplementary Note 1 and Fig. 2b). For instance, the spatiotemporal information of the video playing 'A-P-P-L-E' (Fig. 2c) determines the classification vectors of the final output frame ($h_5$), as shown in the first column of Fig. 2d. The videos playing the other words contain different spatiotemporal information. Because the output currents recorded during the projection of the last letter 'E' ($\Delta I_5$) do not only depend on the present illumination state but also on the number and timing

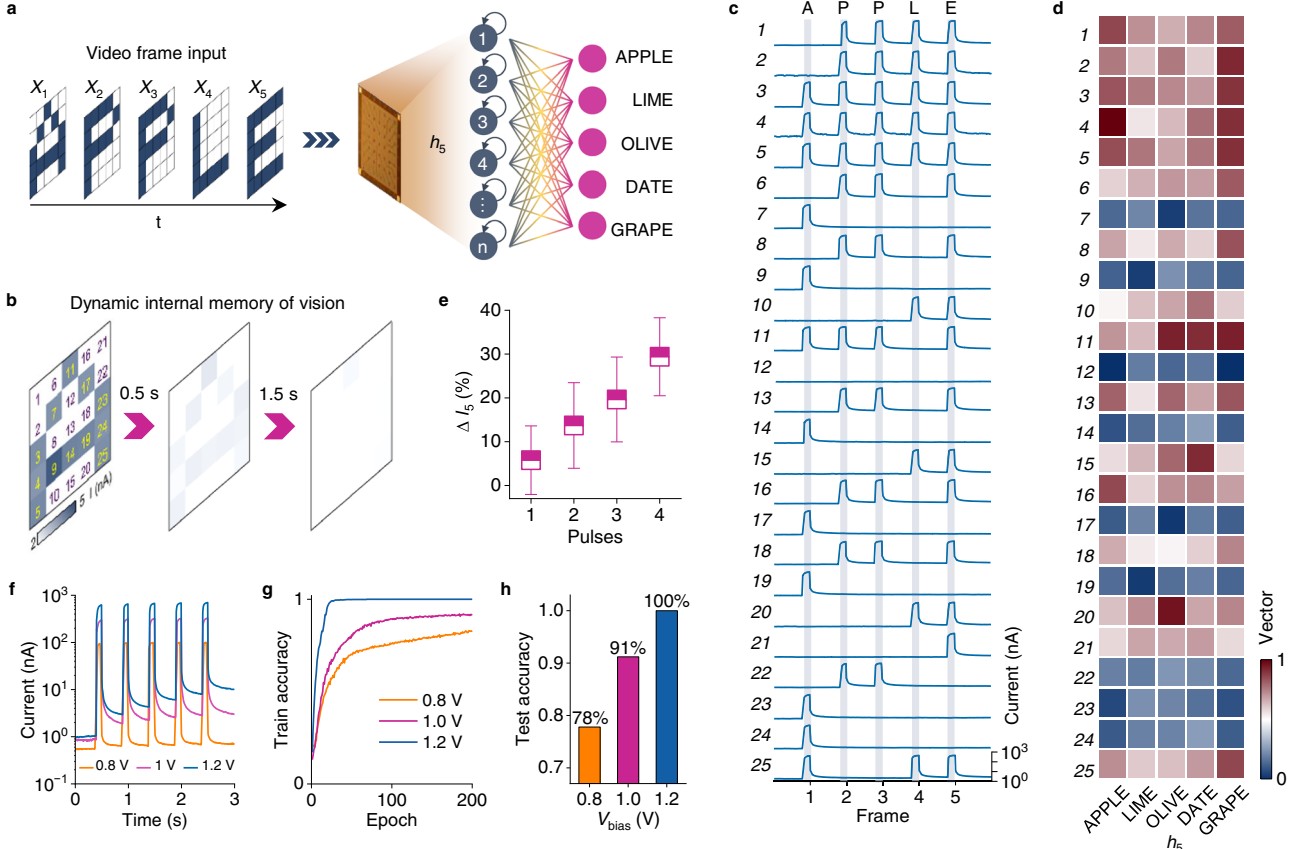

**Fig. 2 | Memory-dependent dynamic vision recognition. a** Videos playing 'APPLE', 'LIME', 'OLIVE', 'DATE', and 'GRAPE' letter-by-letter, all ending with 'E', are used as input to the retinomorphic PMA. Only the photomemristor currents of the last frame ($h_5$) recorded after playing the letter 'E' are used as vectors for recognition by the readout network. **b** Example of how the PMA memorizes the letter 'A'. After an illumination time of 100 ms, the letter 'A' fades in 2 s. **c** Output currents of the 25 photomemristors in the PMA recorded while playing the word 'A-P-P-L-E' letter-by-letter. The light pulses illuminate the PMA for 100 ms, the frame-to-frame rate is 2 Hz, and the bias voltage is 1 V. **d** Feature vectors of the last frame (letter 'E') for the five videos. The vectors are obtained by normalizing the photomemristor currents that are recorded after the 5th frame in (**c**). **e** Change in the photomemristor output current of the last frame as a function of the number of previously received optical pulses. The error bars represent the standard deviation. **f** Photomemristor output current measured at different bias voltages ($V_{bias}$ = 0.8 V, 1.0 V, and 1.2 V) for five optical pulses with a duration of 100 ms and a repetition rate of 2 Hz. The memory states (currents between optical pulses) increase with bias voltage. **g, h** Training and test accuracy for datasets recorded with different bias voltages in the same video classification task. The accuracy increases from 82% for training and 78% for testing at $V_{bias}$ = 0.8 V to 100% for both training and testing at $V_{bias}$ = 1.2 V.

of previously received optical inputs (Fig. 2e), the PMA current map (Supplementary Figs. 7 and 8) and vectors of $h_5$ (Fig. 2d) are unique for each video. The features of the last frame that are read out by the artificial neural network (ANN) thus combine information of the last letter 'E' (Supplementary Fig. 9) and all other letters. Utilizing the spatiotemporal differences, we trained the readout ANN to classify the videos. The ANN has 25 inputs corresponding to the 25 photomemristors of the PMA, and 5 outputs to classify the videos. The recognition accuracies are 97.3% and 91.3% (Supplementary Fig. 10a–d) after 200 training epochs with Gaussian noise factors of $\sigma = 0.15$ and $\sigma = 0.30$, respectively. The confusion matrices show ~100% and ~90% test accuracy with $\sigma = 0.15$ and $\sigma = 0.30$ (Supplementary Fig. 10a–d), enabling accurate DMV tasks. To underline the key role of inherent dynamic memory in dynamic data processing by the RP-RC system, we operated the same PMA without hidden states using the same readout network structure to recognize the videos. In this conventional sensing mode, the readout network uses the peak values of the PMA photoresponse (see constant peaks in Fig. 1d) rather than the memristive states (variable output currents between optical inputs in Fig. 1d). The recognition accuracy achieved using conventional image sensing is just 36.2% for $\sigma = 0.30$ after 200 training epochs (Supplementary Fig. 10e, f), i.e., much lower than the 91.3% attained with the use of hidden states. This

comparison demonstrates the importance and potential of the inherent dynamic memory in the PMA for efficient dynamic vision processing.

In the brain, deep memory usually results in better perception. To evaluate the relation between memory and recognition accuracy in our RP-RC system, we tuned the inherent memory of the PMA by applying different bias voltages ($V_{bias}$ = 0.8, 1.0, and 1.2 V) across the photomemristor Schottky junctions (Fig. 2f). Results for the same videos are shown in Fig. 2g and h and Supplementary Fig. 11. The data demonstrate that the recognition accuracy increases from 78% at 0.8 V to 100% at 1.2 V (Fig. 2h and Supplementary Fig. 11c) because of increased memory of previous frames (Fig. 2f and Supplementary Fig. 11b). This memory-dependent dynamic recognition, which resembles memory-dependent perception in the brain, could enable intelligent sensors with tunable attention.

## Motion recognition and prediction (MRP)

Besides video-of-words recognition, the capabilities of the RP-RC system with hidden memory states can be extended to motion recognition. To demonstrate this, we played three frames representing the motion of an object (a simulated person) at different speeds (slow, medium, fast) (Supplementary Fig. 12). For motion recognition, we used the same PMA and a simple readout ANN with 25 inputs and 3

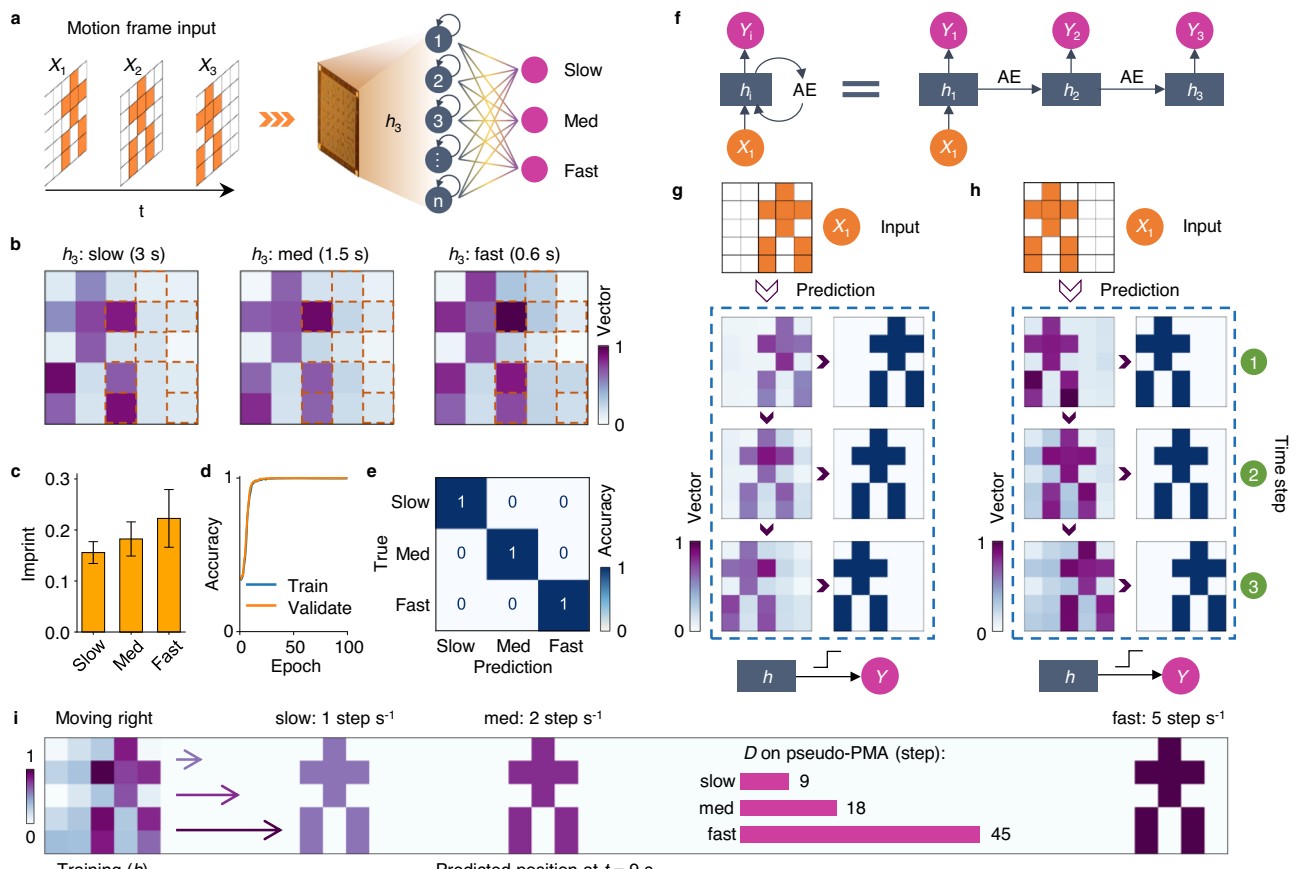

**Fig. 3 | Motion recognition and prediction. a** Programmed light pulses representing object motion are used as input to the retinomorphic PMA. The motion consists of three frames (illumination time of 50 ms per frame), and it is played at three speeds; slow (3 s for 3 frames), medium (1.5 s), and fast (0.6 s). Only the photomemristor output currents recorded after playing the last frame ($h_3$) are used as features for recognition by the readout network. **b** Feature vectors of hidden memory states ($h_3$) of the three motions (slow, medium, fast) obtained after playing the last frame with $V_{bias}$ = 1.0 V. The last frame is recognized for all speeds. Moreover, $h_3$ contains hidden states representing imprints of previous frames and the speed at which they were played. **c** Imprint of previous object positions for different speeds of motion. The imprint factors are calculated by averaging the normalized memory of pixels #9,10,11,13,16,17,18,19,20,22,24,25 (same indices as in

Fig. 2b) in **b**. The selected pixels are all illuminated by optical pulses during object motion. High-speed results in a stronger imprint because of the rate dependence of hidden memory states. The error bars represent the standard deviation. **d** Training and validation accuracy of the readout network. **e** Confusion matrix of motion speed recognition, showing 100% test accuracy. **f** Structure of the RP-RC system for motion prediction. $X_1$ indicates the first frame for prediction, AE is the autoencoder network, and $Y_i$ is the predicted output frame. **g, h** Prediction of an object moving to the left (**g**) and moving to the right (**h**) with input $X_1$. The predicted output frame of the autoencoder, $h$, contains imprints of previous frames as hidden states. The final output $Y$ is obtained by applying a step function to $h$. **i** Predicted position and moved distance ($D$) at $t$ = 9 s following the detection of an object moving to the right at three different speeds.

outputs (Fig. 3a). The feature vectors of the last frame ($h_3$) are different for the three moving speeds because of differences in the hidden memory states (Fig. 3b). Normally, it is impossible to recognize the direction of motion for a completely symmetric object using just one frame. However, in our system, owing to the accumulative photomemristive effect, the imprint of dynamic memory states from previous object positions enables the correct prediction of future trajectories. Moreover, the imprint is stronger when the object moves fast because of the accumulative dynamic photomemristive response (Fig. 3c and Supplementary Fig. 13). Utilizing the spatiotemporal differences in motion imprint, we trained the readout ANN to recognize the motion speed. After 100 training epochs, the training accuracy reaches 100% ($\sigma$ = 0.15) and 97% ($\sigma$ = 0.3) (Fig. 3d and Supplementary Fig. 14), with ~100% test accuracy of the motion speed (Fig. 3e and Supplementary Fig. 14c).

Motion prediction is a crucial task in DMV[4,5]. Inspired by biological spatiotemporal vision prediction[24], our RP-RC system uses learned inherent spatiotemporal sequences in an autoencoder network (Supplementary Fig. 15a) to encode the present vision frame and accurately predict the future frames of moving objects (Fig. 3f).

The autoencoder has 25 inputs corresponding to the 25 photomemristors of the PMA, 10 hidden representations associating input and prediction, and 25 outputs corresponding to the pixels of predicted future frames ($h_{t+1}$, $h_{t+2}$, etc.) with the same dimension as the input frame ($h_t$). When the PMA detects a motion, the system uses the hidden states of the first three motion frames ($h_1$, $h_2$, $h_3$) to train the autoencoder by using $h_1$ and $h_2$, and $h_2$ and $h_3$ as the input and output, respectively. After training the autoencoder with two motions (a symmetric simulated person moving to the left or moving to the right), the RP-RC system successfully predicts the future frames of motion with the first frame as input ($X_1$) (Fig. 3g, h). Under optical input $X_1$, the PMA produces the vectors of the first frame $h_1$. With $h_1$ as autoencoder input, the system successfully predicts $h_2$, and then $h_3$ using $h_2$ as the recurrent input. The final output $Y$ (right columns in Fig. 3g, h) is generated by applying a step function to the hidden memory $h_i$, demonstrating the correct prediction of the two motions. To extend the field of view of the prediction, we introduced a shifting operation to simulate eyeball movement or head rotation towards the target[25]. Continuous prediction of motion enabled by this feature is demonstrated in Supplementary Fig. 15b. Moreover, by

combining motion speed recognition and trajectory prediction, the RP-RC system correctly predicts the exact positions of objects that move with different speeds and into different directions, which is important in traffic assessment and other significant DMV applications. As shown in Supplementary Fig. 16, the readout network first recognizes the speed (time per step) of a moving object using the present frame (as in Fig. 3a–e) and the trained autoencoder predicts the trajectories at a certain future time. From this, the position of the moving object can be calculated accurately. For example, in Fig. 3i, the position of a moving person is predicted 9 s into the future by recognizing the moving speed and predicting the number of steps taken (9 steps for slow, 18 steps for medium, 45 steps for fast motion). By combining the steps and trajectories, accurate positions at any time can be predicted (Supplementary Fig. 16). Prediction results for a completely symmetric object moving right/left at different speeds are shown in Supplementary Movies 1 and 2.

## Intelligent traffic simulation

Safety and efficiency are the most important factors in future intelligent and autonomous traffic, requiring dynamic and accurate recognition and prediction. To demonstrate the potential application of the RP-RC system for future intelligent traffic, we simulated a situation where a robot and a car, both equipped with our RP-RC system, meet at a crosswalk (Fig. 4a). In the simulation, the RP-RC system consists of a PMA with $48 \times 48$ photomemristors, a convolutional neural network (CNN) for speed recognition, and a convolutional autoencoder (CAE) for trajectory prediction (Fig. 4b). As shown in Fig. 4a, W and L mark the width and length of the crosswalk, $x_{car}$ indicates the distance between

the car and the crosswalk at $t = 0$, and $x_{robot}$ specifies the distance between the robot and the crosswalk at the same time. Trained by past motion frames using the dynamic memory of the PMA, the CNN recognizes the object speeds ($v_{car}$ and $v_{robot}$) from the present frames (Supplementary Fig. 17) with >90% average test accuracy (Supplementary Fig. 18) and the PMA-trained CAE (Supplementary Fig. 19) correctly predicts future motion trajectories of the robot and the car, as shown in the upper and lower dashed squares of Fig. 4a. Prediction results of a car and a robot moving right and left at different speeds are shown in Supplementary Movies 3 and 4, demonstrating motion recognition and prediction capabilities for accurate position prediction over a long period of time. If both the car and robot approach the crosswalk, in the car vision (Fig. 4c), the decision (slow down or keep speed) made at $t = 0$ depends on the predicted position of the robot when the car arrives at the crosswalk ($t = x_{car}/v_{car}$). Considering both safety and efficiency, if the robot at $t = x_{car}/v_{car}$ is predicted to be on the crosswalk (dashed square in Fig. 4c), the car slows down. Similarly, in the robot vision, the robot will either slow down or keep its speed based on the predicted position of the car at $t = x_{robot}/v_{robot}$ (Fig. 4d). Based on these safety rules, visual decision maps for the car and the robot can be calculated (Fig. 4e, f and Supplementary Note 2). In the maps, the orange and blue parts indicating 'slow down' and 'keep speed' decisions depend on the car and robot parameters recognized by the RP-RC systems and the width and length of the crosswalk. Moreover, the decision maps are dynamic, as the variables $x_{car}$, $v_{car}$, $x_{robot}$, $v_{robot}$ are changing in real-time. Examples of dynamic decision-making by the car and robot are shown in Fig. 4g, h. The car or robot slows down when the pink marker indicating their position is in the yellow area and they retain their speed when the marker enters the blue

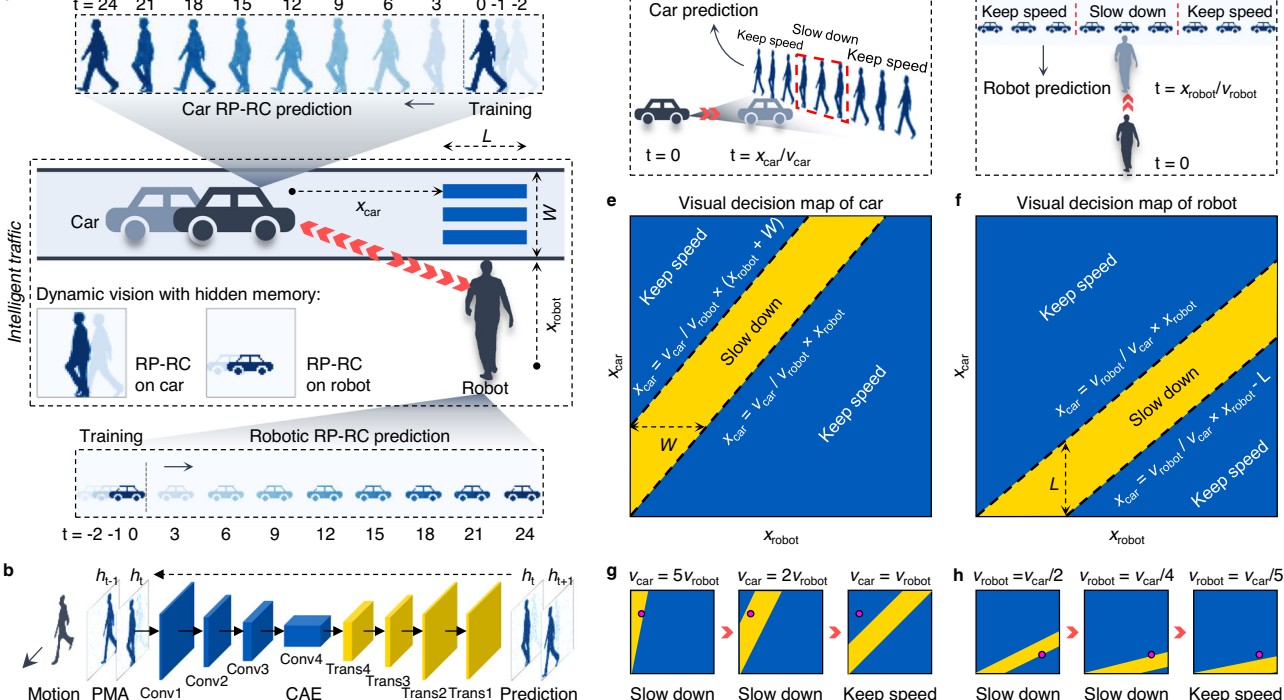

**Fig. 4 | Intelligent traffic simulation with RP-RC systems. a** Simulation of a robot and a car equipped with an RP-RC system. Imprint values (Fig. 3c) are used to generate the first three frames of the PMA with hidden memory. After training by the first three frames of a moving robot and a moving car, the RP-RC systems predict future motion trajectories for both objects based on a single vision input. The first 24 steps of the predicted motions are demonstrated. $x_{car}$ and $x_{robot}$ indicate the distance between the car and the robot to the crosswalk. W and L are the width and length of the crosswalk. **b** Structure of the convolutional autoencoder (CAE) used in the RP-RC systems. **c** Motion prediction by the RP-RC system for the car. The car will slow down at $t = 0$ if the robot is predicted to be on the crosswalk at $t = x_{car}/v_{car}$. Otherwise, the car will continue its motion at the same speed. **d** Motion prediction by the RP-RC system for the robot. The robot will slow down at $t = 0$ if the car is predicted to be on the crosswalk at $t = x_{robot}/v_{robot}$. Otherwise, the robot will continue its motion at the same speed. **e, f** RP-RC system decision maps of the car and the robot. Orange and blue areas indicate that the car and robot need to slow down or can keep their speed. **g, h** Two examples of dynamic decision-making by the car and robot at different initial positions based on motion recognition and prediction.

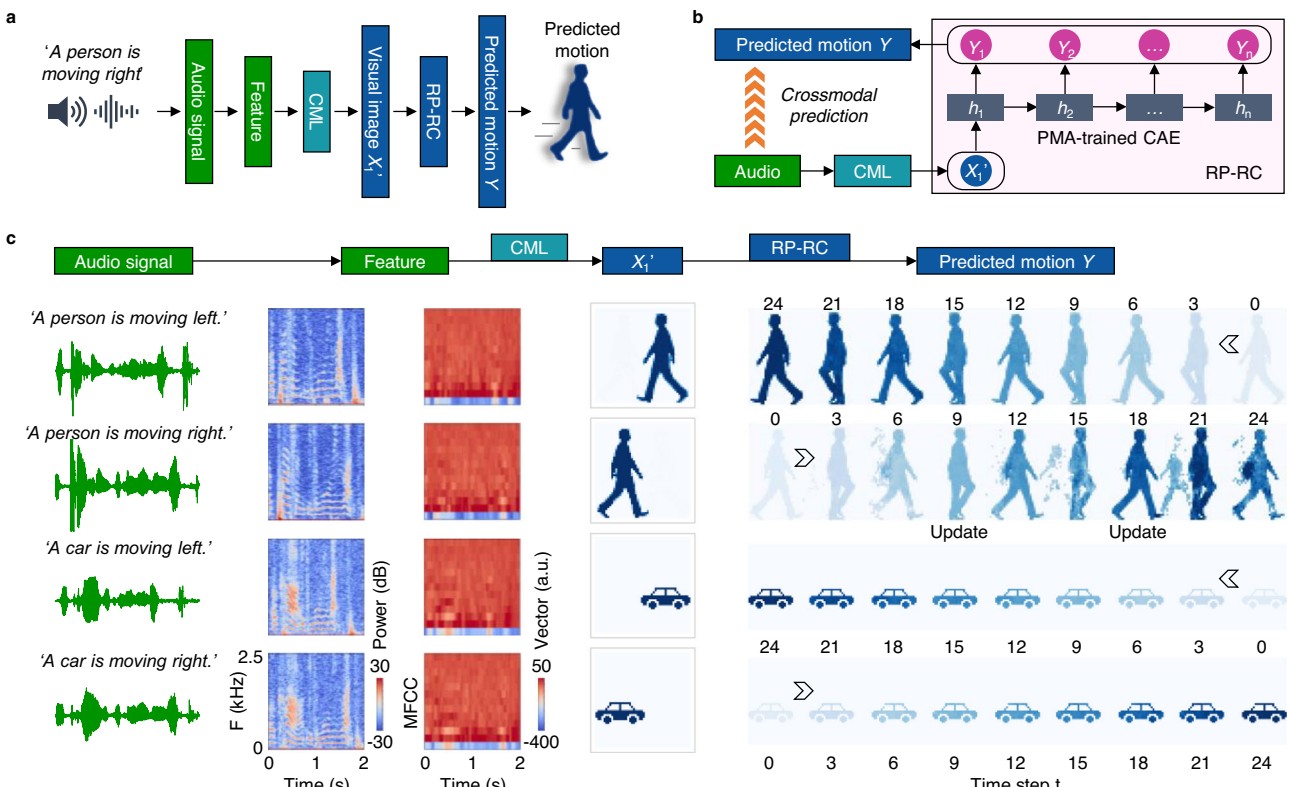

**Fig. 5 | Crossmodal audio-to-visual motion prediction with the RP-RC system. a** Schematic of the audio-to-motion data flow, including audio feature extraction, crossmodal learning (CML), and recurrent vision prediction. **b** Schematic of the crossmodal audio-to-motion prediction system. The PMA-trained CAE is identical to the one used in Fig. 4. **c** Audio-to-visual motion prediction results. Audio signals '*A person is moving right*', '*A person is moving left*', '*A car is moving right*', '*A car is moving left*' are used as input. For each input, Mel spectrograms and MFCC features are extracted for crossmodal recognition of the first motion frame ($X_1'$) for crossmodal recognition by a trained DNN. The recognition accuracy of $X_1'$ through audio input is 90% for 120 random test datasets (Supplementary Fig. 20). Audio-to-motion prediction is successful for three of the four motions (the first 25 predicted frames are shown). The motion of '*A person is moving right*' is predicted successfully for the first 9 frames but then fades (Supplementary Fig. 21). By giving another audio input at step 9, the correct prediction of motion is re-established with weaker residual imprints from the previous frames.

area of the dynamic decision map. Dynamic visual decision-making based on MRP by the RP-RC system, as demonstrated here, is relevant for compact DMV applications in future hybrid or autonomous intelligent traffic.

In the brain, cortices process multimodal environmental information by crossmodal association. For instance, audio-to-motion prediction plays a vital role in a person's ability to anticipate danger and in the communication between dolphins[26] (https://cymascope.com/oceanography/). To emulate crossmodal prediction using a single compressed frame, we associated motion recognition and prediction in our RP-RC system with audio inputs through crossmodal learning (Fig. 5a). In this scheme, the system converts audio signals to Mel-frequency cepstral coefficients (MFCC) features and a deep neural network (DNN) processes the features to crossmodally recognize the first frame of motion ($X_1'$). Feeding this first frame into a PMA-trained CAE (see also Fig. 4) then provides the prediction of continuous motion, as illustrated in Fig. 5b. As an example, we demonstrate audio-to-motion prediction of a moving car or person. The audio input signals are '*A person is moving right*', '*A person is moving left*', '*A car is moving right*', and '*A car is moving left*' ('Audio signal' in Fig. 5c). The MFCC features corresponding to these audio inputs ('Feature' in Fig. 5c) train the DNN, leading to crossmodal recognition of the first motion frame ($X_1'$ in Fig. 5c and Supplementary Fig. 20). Next, the PMA-trained CAE already employed in Fig. 4 encodes the recognized vision frame and predicts the next frame. The predicted frame is then fed back to the CAE to predict the next frame, etc., thus realizing continuous crossmodal

audio-to-motion prediction (Predicted motion *Y* in Fig. 5c and Supplementary Fig. 21). Conceptual crossmodal motion prediction will further enhance robotic dynamic vision for human-machine traffic communication, autonomous driving, and intelligent transport technology.

## Discussion
We have demonstrated a retinomorphic photomemristor-reservoir computing (RP-RC) system consisting of a photomemristor array (PMA) and readout networks. The inherent dynamic memory of the PMA compresses spatiotemporal vision information into a single frame, eliminating redundant data flow and facilitating various dynamic visual tasks. Spatiotemporal analysis is important for dynamic information processing. In previous research, ionic memristors[27] and phase-change memtransistors[28] with tunable plasticity have emulated the dynamics of synapses for temporal data classification and forecasting. These reservoir computing systems detect the temporal data streams externally. Traditional vision sensing systems analyze dynamic vision using multiple frames with intra-frame compression, which requires frequent data transmission between separate sensing, memory, and processing modules. Our RP-RC system senses, memorizes, and processes optical-image sequences with a single PMA. This compact in-sensor memory and computing solution exploits a photomemristive effect with short-term memory for in-frame/photomemristor compression, facilitating motion recognition and prediction, and power efficiency. Furthermore, because the bias voltage across the Schottky junctions

tunes the photomemristor output currents, the sensing and memory functions are adjustable for speed, accuracy, or other requirements of intelligent in-sensor processing tasks. Additionally, our recurrent photomemristor networks can be easily extended to other modalities via crossmodal learning, fulfilling the requirement of applications in multimodal environments. Although the 3.1 eV bandgap of ZnO limits the photoresponse range of the current photomemristor to 320–400 nm (UV–blue light), broadening of this response to the visible range is possible by doping ZnO with Co[29] or using other materials with narrower bandgaps. A comparison of our photomemristor-based system with other reservoir computing systems and traditional vision sensors is given in Supplementary Tables 1 and 2. Above all, the demonstrated recurrent photomemristor networks hold great potential for urgent dynamic machine vision (DMV) applications requiring accurate on-site motion perception and prediction.

## Methods

### Fabrication of photomemristor array (PMA)
The PMA consisted of indium tin oxide (ITO)/ZnO/Nb-doped SrTiO$_3$ (NSTO) junctions fabricated by atomic-layer deposition (ALD), photolithography, etching, and magnetron sputtering. Conductive NSTO substrates were used as the bottom electrode of the photomemristors. To form a Schottky barrier, photosensitive ZnO films with a thickness of 60 nm were deposited by magnetron sputtering (5.8 × 10$^{-3}$ mbar, Ar 16 sccm, O 4 sccm, power 60 W) on top of the NSTO substrates. Transparent and conductive ITO top electrodes were grown by magnetron sputtering (3.4 × 10$^{-3}$ mbar, Ar 10 sccm, power 50 W). The photomemristors had a working area of 100 μm × 100 μm. Besides the working area, an insulation Al$_2$O$_3$ layer was deposited by ALD between the ZnO film and the ITO electrode wires and pads. The working areas were opened by wet etching.

### Characterization of photomemristor array (PMA)
The photomemristor array was measured using an electrical measurement system consisting of a Keithley 4200 semiconductor characterization unit, a Keithley 2400 source meter, an Agilent B1500A semiconductor device parameter analyzer, a Tektronix AFG 1062 arbitrary function generator, a Keysight DSO1024A oscilloscope, and a blue LED with shadow masks functioning as the optical input. The intensity of the blue light pulses was 0.65 ± 0.06 mW mm$^{-2}$, which was calibrated by a Thorlabs FD11A photodetector and an Ocean Optics USB2000 + optical spectrometer. Programmed light pulses were used to simulate image and optical motion sequence inputs. The current map of the PMA was recorded by an Agilent B1500A semiconductor device parameter analyzer one-by-one under one-by-one optical input. The bias voltages were applied on the ITO electrodes.

### Readout network for video recognition of words
The word classification network had 25 input neurons corresponding to the 5 × 5 PMA, and 5 output neurons corresponding to 5 classes of videos playing words all ending with the letter 'E'. 1200 datasets (900 datasets for training and 300 datasets for testing) were generated by adding Gaussian noise to the recorded output currents for training. This method was chosen over the modification of input frames (i.e. via pixel flipping[30]) because it is more practical for videos. The noise rates were 15% and 30%. We trained the network with a batch size of 25 and 200 training epochs and achieved an accuracy of 97.3% and 91.3% on the test dataset with noise rates of 15% and 30%, respectively (Supplementary Fig. 10a–d).

### Readout network for video recognition of motion
The motion speed classification network had 25 input neurons corresponding to the 5 × 5 PMA, and 3 output neurons corresponding to three classes of speeds. 1200 datasets (900 datasets for training and 300 datasets for testing) were generated by adding noise to the experimental data with noise rates of 15% and 30%. We trained the network with a batch size of 25 and 100 training epochs and achieved an accuracy of 100% and 97% on the test dataset with noise rates of 15% and 30%, respectively (Fig. 3d and Supplementary Fig. 14).

### Autoencoder network for motion prediction
An autoencoder network was used for motion prediction. The encoder part had 25 input neurons and 10 output neurons, and the decoder part had 10 input neurons and 25 output neurons (Supplementary Fig. 15a). The activation functions were softmax and sigmoid for the encoder and decoder. The loss function was a mean squared error function. 96,000 datasets (72,000 for training and 24,000 for testing) were generated by adding noise to the experimental data with a noise rate of 15%. We trained the autoencoder with a batch size of 100 and 100 training epochs. After training, the RP-RC system successfully predicted the motions with the first frame as input (Fig. 3g, h).

### Convolutional neural network (CNN) for motion speed classification
A CNN was used to classify the motion speeds of a robot and a car. The CNN had 4 Conv2D layers and 4 MaxPooling2D layers to extract the features in the present frame ($h_3$) with the memory of previous frames. A fully connected layer was used to classify the features. 16,800 datasets (12,000 for training, 2400 for validating, and 2400 for testing) were generated with a noise factor of 10%. We trained the CNN with a batch size of 100 and 100 training epochs. More details can be found in Supplementary Figs. S17 and S18.

### Convolutional autoencoder (CAE) for motion prediction
To implement complex motion prediction (person and car motion), a convolutional autoencoder (CAE)[31] was used to predict motion frames. The CAE is an unsupervised neural network model for feature extraction of images. The input and output frames had 48 × 48 pixels (Supplementary Fig. 19). The CAE had 4 Conv2D layers and 4 MaxPooling2D layers at the encoding side, and 4 Conv2DTranspose layers at the decoding side (Fig. 4b). 64,000 datasets (32,000 for training and 32,000 for testing) were generated by adding noise to simulated motions (Supplementary Fig. 19) with a noise rate of 10%. We trained the CAE with a batch size of 160 and 400 training epochs. After training, the CAE successfully predicted the frames of motion. (Fig. 4a).

### Deep neural network (DNN) for crossmodal learning (CML)
A DNN was used for audio recognition and the generation of the first frame of motion (Fig. 5 and Supplementary Fig. 20). The DNN had 52 input neurons corresponding to Mel features of audio signals[32], 25 neurons in the first hidden layer, 15 neurons in the second hidden layer, and 2304 output neurons corresponding to 48 × 48 pixels of the first frame in the simulated motion. The activation functions were relu and sigmoid for the hidden and output layers. The loss function was a mean squared error function. 97,920 datasets of Mel features (81,600 for training and 16,320 for testing) were generated by adding noise to the experimental audio data with a noise of 10%. We trained the DNN with a batch size of 200 and 150 training epochs. After training, the DNN successfully predicted the first frame of motion ($X_1$) upon audio input (Supplementary Fig. 20) with a test accuracy of ~90%. Then the recognized $X_1$ frame was used as input to the trained convolutional autoencoder (CAE) in Fig. 4 for continuous motion prediction (Fig. 5 and Supplementary Fig. 21).

All the algorithms were written in Python on the TensorFlow platform.

**Reporting summary**

Further information on research design is available in the Nature Portfolio Reporting Summary linked to this article.

## Data availability

The source data underlying the figures in the main manuscript and Supplementary Information are provided as Source Data file. The data that support the findings of this study are available from the corresponding authors upon reasonable request. Source data are provided with this paper.

## Code availability

The code that supports the results within this paper and the other findings of this study are available from the corresponding authors upon reasonable request.

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

## Acknowledgements

We gratefully acknowledge E.I. Kauppinen for providing infrastructure support for the electrical measurements. H. Tan. thanks, R. He for inspiration and discussion on the main concept. We acknowledge H. Qin and Y. Zhou for their fruitful discussions and contributions to coding. The project made use of the OtaNano—Micronova Nanofabrication Center and the OtaNano—Nanomicroscopy Center, supported by Aalto University. This work was supported by the Academy of Finland (Grant no. 316973 H. T. and 13293916 S.v.D.).

## Author contributions

H.T. and S.v.D. initiated the research. H.T. conceived the idea and designed the systems. H.T. fabricated the PMA. H.T. conducted the electrical and optoelectronic measurements. H.T. wrote the code of the machine learning algorithms. H.T. and S.v.D. analyzed the data. H.T. and S.v.D. wrote the manuscript. All authors discussed the results and commented on the manuscript.

## Competing interests

The authors declare no competing interests.
