## [Peer Review File · Nature Communications]

REVIEWER COMMENTS

Reviewer #1 (Remarks to the Author):

I have no further question on this paper.

Reviewer #2 (Remarks to the Author):

In this manuscript, the authors demonstrate a retinomorphic photomemristor system for smart vision, here referred as dynamic machine vision. By taking advantage of the history-dependent response of the photomemristor array, the system is able to recognize frame sequences comprising "moving objects". Finally, in a simulated scenario, it is shown that such concept can be used for trajectory/traffic prediction and adjustment in future autonomous agents. The manuscript is well-organized and the whole concept is novel. I therefore, believe that it is highly suitable for publication in *Nature Communications*. I only have a few comments in order to strengthen the manuscript.

1. Relevant manuscripts, such as *Nature Nanotechnology* 17, 507–513 (2022), should be discussed. The difference should be clear.
2. A minimum frame-to-frame time of 0.6s is experimentally demonstrated that corresponds to a framerate of ~1.6fps. The authors should discuss ways of improving or limitations towards "natural" framerates (~25 fps). A discussion around materials and devices would be appropriate.
3. A table of metrics (speed, compression, energy, etc.) showcasing the advantages of this approach against traditional approaches for vision would be beneficial.

4. Other modalities, such as optical frequency dependent absorption could also be discussed (in terms of applications).

Reviewer #3 (Remarks to the Author):

In my opinion, the claim made in this manuscript should be treated with caution. It is correct to say that the photomemristor array can process a moving scene in-situ due to its dynamic memory property. However, it is misleading and inaccurate to state that recognition is accomplished within a single frame. To reach an inference, the photomemristor array must already have processed numerous previous frames and generate a unique output that can be classified reliably. Except for the fact that this processing occurs in an analogue manner within the sensor without having the data to be first digitized and then moved to memory, the process is no different from that of a frame-based machine vision system. Works based on CNN have already shown that the technique may be extended to account for spatiotemporal input. The advantage offered by physical reservoir computing is that only the readout layer needs to be trained since the mapping is inherently determined by the internal dynamics of the devices. It therefore provides a resource-lean approach towards the management of big data. There are already many papers that studied how physical reservoir computing can be applied for motion recognition.

In view of the above doubt on the accuracy of the claim made, I do not recommend the publication of this manuscript.

Reviewer #4 (Remarks to the Author):

In this work by Tan et al., a retinomorph in-sensor reservoir computing system is demonstrated based on a photomemristor array, which exhibits light-induced short-term memory with an accumulative effect. The 'imprint' of previous states serves as the basis of realizing various dynamic machine vision (DMV) tasks, such as single-frame dynamic vision recognition, single-frame motion recognition and prediction, and intelligent traffic simulation. This is an interesting work combining reservoir computing with DMV applications. However, I have the following questions for the preparation of a revised manuscript.

1. In the single-frame video recognition task, the authors compared the PMA with or without hidden states (previous memory), and the former achieved a much higher recognition accuracy. What I'm

curious about is whether more remaining previous memory results in better accuracy. Can authors clarify this? Also, how are the normalized states in Fig. 2d obtained? Is it by normalization of the device current (I) or current change (ΔI)?

2. The training and testing datasets in Fig. 2 are generated by adding noise to the experimental acquired current states rather than adding noise to the input pictures (e.g. <https://doi.org/10.1038/s41467-017-02337-y>). Could the authors discuss the effects of these two approaches?

3. The authors state that 'Moreover, the uniform photomemristive response (Supplementary Fig. 3 and 4) guarantees high accuracy in various DMV tasks.' It is nice to see the photomemristors with uniform responses, but for the single-frame dynamic vision recognition task, will the inhomogeneity of the device (D2D variability) necessarily reduce the accuracy?

4. The imprint factors in Fig. 3c are extracted from Fig. 3b. Could the authors give more detailed descriptions how the imprint factors are extracted? The imprint factor seems to be the bridge between experiments and simulations for the task in Fig. 4, and should be clearly elaborated.

In the point-by-point response:

1. Reviewer comments are in black.
2. Our response is in blue.
3. Changes to the manuscript and the Supplementary Information are in red.

Reviewer #1:

General comment

I have no further question on this paper.

Our reply

We thank the reviewer for the comment on our manuscript.

Reviewer #2:

General comment

In this manuscript, the authors demonstrate a retinomorphic photomemristor system for smart vision, here referred as dynamic machine vision. By taking advantage of the history-dependent response of the photomemristor array, the system is able to recognize frame sequences comprising "moving objects". Finally, in a simulated scenario, it is shown that such concept can be used for trajectory/traffic prediction and adjustment in future autonomous agents. The manuscript is well-organized and the whole concept is novel. I therefore, believe that it is highly suitable for publication in *Nature Communications*. I only have a few comments in order to strengthen the manuscript.

Our reply

We thank the reviewer for the positive assessment of our manuscript and his/her strong recommendation for publication. We also appreciate the reviewer's suggestions to strengthen the manuscript. Below we address the comments on a point-by-point basis.

Reviewer comment 1

Relevant manuscripts, such as *Nature Nanotechnology* **17**, 507–513 (2022), should be discussed. The difference should be clear.

Our reply

The article in '*Nature Nanotechnology* 17, 507–513 (2022)' reports on phase-change memtransistors, which utilize both non-volatility and volatility for the emulation of tunable plasticity (STP and LTP). The authors demonstrate memtransistor-based accelerators for Hopfield neural networks and use their system to solve combinatorial optimization problems and recognize motion. The work clearly showcases the efficacy of memtransistors with tunable dynamic plasticity for neuromorphic computing.

In our manuscript, we report on photomemristors, which not only provide dynamic memory, but also optical sensing without the need for external sensors. The dynamic memory of detected optical information is used to directly integrate multiple previous frames of the detected dynamic vision into a single informative present frame (in-frame or in-

photomemristor hardware compression method), with which the detected dynamic vision can be accurately recognized or predicted via trained readout networks. The in-frame or in-photomemristor compression doesn't require any additional signal transmission or conversion, thus reducing power and time consumption. Our work clearly demonstrates the potential of photomemristors for retinomorphic dynamic vision sensors. In short, the most important difference is that our photomemristors do not only provide dynamic processing, but also integrate optical sensing capabilities without the need for additional sensors, enabling in-photomemristor sensing and dynamic processing, and providing a novel approach for intelligent dynamic sensing technology. Besides, there are differences in operation mechanism, device structure, energy consumption, and application area. The differences between the two works and other physical reservoir computing systems are summarized in a revised Supplementary Table 1 (see below).

Supplementary Table 1. Comparison of our photomemristor with typical reported physical reservoir computing systems.

Structure	Mechanism	Sensing	On/off	Power/Energy per operation [§]	Dynamic processing	Classification	Prediction	Ref.
FeB/MgO/CoFeB	Spin-torque nano-oscillator	-	-	-	In-memory	Spoken-digit, waveform	-	3
W/WO ₂ /Pd	Ion migration	-	~ 10	3 nJ	In-memory	Spoken-digit	Mackey-Glass time series	4
Ge ₁₅ Sb ₈₅ -based	Phase change	-	~ 10 ²	< 100 nW	In-memory	Motion	-	5
Au/Cr/SnS/Cr/Au	Photogating	✓	~ 2	85 nJ	In-sensor	Language	-	6
Au/P(VDF-TrFE)/Cs ₂ AgBiBr ₆ /ITO	Photo-generated carriers pinning	✓	-	Self-powered	In-sensor	Image, vehicle flow	-	7
ITO/ZnO/NSTO	Optoelectronic Schottky barrier	✓	~ 10 ²	15 nJ [#] 30 nJ [*]	In-sensor or In-photomemristor	Language, Single-frame MRP	-	This work

SUPPLEMENTARY REFERENCES

5. Sarwat, S. G., Kersting, B., Moraitis, T., Jonnalagadda, V. P. & Sebastian, A. Phase-change memtransistive synapses for mixed-plasticity neural computations. *Nat. Nanotechnol.* **17**, 507–513 (2022).

Moreover, the following text with references is now included on page 12 of the revised manuscript (Discussion section):

Spatiotemporal analysis is important for dynamic information processing. In previous research, ionic memristor²⁷ and phase-change memtransistor²⁸ with tunable plasticity have emulated the dynamics of synapses for temporal data classification and forecasting. These reservoir computing systems detect the temporal data streams externally. Traditional vision sensing systems analyze dynamic vision using multiple frames with intra-frame compression, which requires frequent data transmission between separate sensing, memory and processing modules. Our RP-RC system senses, memorizes, and processes optical-image sequences with a single PMA. This compact in-sensor memory and computing solution exploits a photomemristive effect with short-term memory for in-frame/photomemristor compression, facilitating single-frame motion recognition and prediction, and power efficiency.

REFERENCES

27. Moon, J., Ma, W., Shin, J. H., Cai, F., Du, C., Lee, S. H., Lu, W. D. Temporal data classification and forecasting using a memristor-based reservoir computing system. *Nat. Electron.* **2**, 480-487 (2019).
28. Sarwat, S. G., Kersting, B., Moraitis, T., Jonnalagadda, V. P. & Sebastian, A. Phase-change memtransistive synapses for mixed-plasticity neural computations. *Nat. Nanotechnol.* **17**, 507–513 (2022).

Reviewer comment 2

A minimum frame-to-frame time of 0.6s is experimentally demonstrated that corresponds to a framerate of ~1.6fps. The authors should discuss ways of improving or limitations towards "natural" framerates (~25 fps). A discussion around materials and devices would be appropriate.

Our reply

We agree with the reviewer that the demonstrated working framerate is lower than 'natural' framerates (~25 fps). However, the framerate of our retinomorphic photomemristor system can easily be extended to 25 fps and beyond.

The working speed of a photomemristor depends on the materials and device structure. Sensing materials with a direct bandgap have a fast photoresponse. The optoelectronic response of the Schottky junction is also fast because of a built-in electric field, which quickly separates the photoinduced carriers. Our photomemristors are made of ITO/ZnO/NSTO. The direct bandgap of ZnO and the Schottky barrier at the ZnO/NSTO interface provide fast optoelectronic sensing. To address the reviewer's comment, we tested the photomemristors with a 60 Hz optical input signal (corresponding to 60 fps), which is faster than the 'natural' framerate. In the new experiments, we used 5 ms optical pulses and we recorded the photocurrents with a bias voltage of 1 V (new Supplementary Fig. 3a). Dynamic memory states (short-term memory) within 9 ms after each pulse were recorded (new Supplementary Fig. 3b-3d). The overall level of the dynamic memory states increases with pulses number. The first five memory states are distinguishable and increasing (new Supplementary Fig. 3e), enabling dynamic vision tasks. We added the following text on page 5 of the revised manuscript:

'To test the speed of photosensing with memory by the PMA, we increased the frequency of the optical input to 60 Hz (Supplementary Fig. 3), corresponding to the frequency of commercial displays. Again, a large continuous range of dynamic analog states is measured up to hundreds of optical pulses, demonstrating adequate in-sensor memory for the processing of hidden states.'

The new Supplementary Figure 3 is shown below.

Supplementary Figure 3. Photomemristive switching behavior at 60 Hz, corresponding to 60 frames per second (fps). **a** Sensing, accumulating, and memory of optical inputs (X) using 5 ms pulses at a repetition rate of 60 Hz and a photomemristor bias voltage of 1.0 V. The photomemristor senses and temporally memorizes optical information through the slowly decaying photomemristor current. **b** More than 400 dynamic analog hidden states of the photomemristor measured after applying repeated optical pulses with a duration of 5 ms and at a repetition rate of 60 Hz. **c, d** Details of the photomemristor response recorded at 3 s and 9 s in **a**. **e** Current memory states for the first five optical pulses, indicating distinguishable and increasing memory with pulse number. The values are extracted from the measurements by averaging the current over 5 ms after each pulse.

Reviewer comment 3

A table of metrics (speed, compression, energy, etc.) showcasing the advantages of this approach against traditional approaches for vision would be beneficial.

Our reply

We added a table of metrics (Supplementary Table 2) comparing traditional approaches and our approach for dynamic vision to the Supplementary Information (see below). Besides, we added relevant references to the Supplementary Information. Finally, we now directly

compare our approach to traditional approaches in the Discussion section of the main manuscript (page 12). Supplementary Table 2 and the new text in the Discussion section are given below.

Supplementary Table 2. Comparison of our photomemristor-based system with traditional vision sensor for dynamic vision processing.

Sensor	Pixel type	Speed	Compression method	Dynamic processing	Storage	Energy per operation [§]	Ref.
Traditional vision sensor	Photodiode	30-240 fps	Inter-frame	Separated processor	Separated memory	Sensing: 0.42 nJ + RC: 143 nJ	8-10
Dynamic photomemristor vision sensor	Photomemristor	1-5 fps [#] 60 fps [*]	In-frame or In-photomemristor	In-photomemristor		1-5 fps: 30 nJ 60 fps: 1.5 nJ	This work

[#]1-5 fps indicates the speed of demonstrated dynamic vision processing in this manuscript.

^{*}60 fps indicates the theoretical speed that was tested achievable (Supplementary Fig. 3).

[§]The energy per operation in photomemristor was calculated by $V \times I \times t$, where V is program (electrical) or reading (optical) voltage, I is the current under V, t is the pulse width. The energy per operation in traditional vision sensing system was calculated by adding energy per operation of sensor (0.42 nJ)⁸ and FPGA-based reservoir computing (RC) (143 nJ)⁹.

SUPPLEMENTARY REFERENCES

8. <https://www.dpreview.com/news/2183540037/samsung-65-14nm-stacked-sensor-design-power-efficiency-density-mobile-image-sensors>
9. Aloma, M. L. et al. Digital Implementation of a single dynamical node reservoir computer. *IEEE Trans. Circ. Syst. II* 62 10, 977–981 (2015).
10. Abomhara, M., Khalifa, O. O., Zakaria, O., Zaidan, A. A. & Rame, A. Video compression techniques: An overview. *J. Appl. Sci.* 10, 1834-1840 (2010).

New text in the Discussion section:

‘Traditional vision sensing systems analyze dynamic vision using multiple frames with intra-frame compression, which requires frequent data transmission between separate sensing, memory, and processing modules. Our RP-RC system senses, memorizes, and processes optical-image sequences with a single PMA. This compact in-sensor memory and computing

solution exploits a photomemristive effect with short-term memory for in-frame/photomemristor compression, facilitating single-frame motion recognition and prediction, and power efficiency. Furthermore, because the bias voltage across the Schottky junctions tunes the photomemristor output currents, the sensing and memory functions are adjustable for speed, accuracy, or other requirements of intelligent in-sensor processing tasks.'

Reviewer comment 4

Other modalities, such as optical frequency dependent absorption could also be discussed (in terms of applications).

Our reply

To check the frequency range of the photomemristors, we conducted UV-Vis absorption measurement using a Cary Series UV-Vis-NIR spectrometer (Agilent Technology). The results shown in Figure R1 demonstrate that ZnO absorbs most light in the UV – blue range (320 – 400 nm), consistent with a direct bandgap of 3.1 eV. For dynamic vision sensing, a good photoresponse to the full visible range is desirable. To implement this, other photomemristor materials with a smaller bandgap should be considered. We plan to explore this in future works. The following options are under consideration: 1) Doping ZnO with Co (*J. Mater. Sci.: Mater. Electron.* 29, 12917–12926 (2018)); 2) Using novel materials with a tunable photoresponse such as monolayer MoS₂-based mem-devices (*Nature* 554, 500–504 (2018) or van der Waal junctions (*Science* 378, 296–299 (2022))).

Figure R1. Optical absorption spectrum of a photomemristor with an ITO/ZnO/NSTO structure.

We added the following text on page 12 (Discussion section) of the revised manuscript:

‘Although the 3.1 eV bandgap of ZnO limits the photoresponse range of the current photomemristor to 320 nm – 400 nm (UV – blue light), broadening of this response to the visible range is possible by doping ZnO with Co²⁹ or using other materials with narrower bandgaps.’

REFERENCES

29. Ji, H., Cai, C., Zhou, S. & Liu, W. Structure, photoluminescence, and magnetic properties of Co-doped ZnO nanoparticles. *J. Mater. Sci.: Mater. Electron.* **29**, 12917–12926 (2018).

Reviewer #3:

General comment

In my opinion, the claim made in this manuscript should be treated with caution. It is correct to say that the photomemristor array can process a moving scene in-situ due to its dynamic memory property. However, it is misleading and inaccurate to state that recognition is accomplished within a single frame. To reach an inference, the photomemristor array must already have processed numerous previous frames and generate a unique output that can be classified reliably. Except for the fact that this processing occurs in an analogue manner within the sensor without having the data to be first digitized and then moved to memory, the process is no different from that of a frame-based machine vision system. Works based on CNN have already shown that the technique may be extended to account for spatiotemporal input. The advantage offered by physical reservoir computing is that only the readout layer needs to be trained since the mapping is inherently determined by the internal dynamics of the devices. It therefore provides a resource-lean approach towards the management of big data. There are already many papers that studied how physical reservoir computing can be applied for motion recognition. In view of the above doubt on the accuracy of the claim made, I do not recommend the publication of this manuscript.

Our reply

We thank the reviewer for evaluating our work. Below we address the reviewer's concern regarding the use of the 'single-frame' terminology and the novelty of our reservoir computing approach.

Single-frame: The photomemristors of our system detect and integrate multiple previous frames into a single present frame through their inherent dynamic memory. Only this single frame is used for further processing and motion recognition/prediction. Dynamic in-frame/in-photomemristor hardware-based compression of motion frames, allowing processing based on a single all-informative frame, has not been demonstrated before. In the revised manuscript, we clearly state that detection requires multiple frames, but that motion recognition and prediction is based on a single frame with embedded memorized information. For instance, on page 4 we now write:

'Here, we for the first-time exploit photomemristors for MRP based on a single frame with embedded memorized information from multiple previous frames.'

Novelty of our reservoir computing approach: While there are many works on physical reservoir computing based on short-term *memory effects* in a dynamic reservoir (see for example *Nature* **547**, 428-431 (2017); *Nat. Electron.* **2**, 480-487 (2019); *Nat. Mater.* **21**, 195–202 (2022)), these implementations do not have *sensing* and *in-sensor processing* capabilities. Our photomemristor approach to in-memory reservoir computing has several advantages: 1) integration of optical sensing, memory, and processing functions in photomemristors; 2) in-sensor reservoir of dynamic vision frames; 3) inherent in-frame/in-photomemristor data compression; 4) compact in-sensor motion recognition and prediction for dynamic machine vision; 5) low power operation compared to traditional dynamic vision sensing and processing. Only a few previous works report on in-sensor reservoir computing (most notably *Sci. Adv.* **7**, eabg1455 (2021); *Adv. Sci.* **9**, 2106092 (2022)). In these studies, language and image recognition were demonstrated using dynamic processing of visual information. However, dynamic motion recognition and prediction with integrated and compact retinomorph sensors has not been reported. In our work, we for the first-time exploit photomemristor arrays that do not only sense visual information, but also provide inherent memory for the embedding of multiple previous frames into a single present frame. This capability, which originates in the photomemristive effect, facilitates in-sensor motion recognition and prediction. We provide a direct comparison of our retinomorph photomemristor-reservoir computing (RP-RC) system and other physical reservoir computing approaches in Supplementary Table 1. Additionally, Supplementary Table 2 compares our dynamic photomemristor vision sensor to traditional vision sensors. We also discuss the novelty of our work in relation to other works in the revised text on page 11-12 of the main manuscript. The new text reads:

‘Spatiotemporal analysis is important for dynamic information processing. In previous research, ionic memristor²⁷ and phase-change memtransistor²⁸ with tunable plasticity have emulated the dynamics of synapses for temporal data classification and forecasting. These reservoir computing systems detect the temporal data streams externally. Traditional vision sensing systems analyze dynamic vision using multiple frames with intra-frame compression, which requires frequent data transmission between separate sensing, memory, and processing modules. Our RP-RC system senses, memorizes, and processes optical-image sequences with a single PMA. This compact in-sensor memory and computing solution

exploits a photomemristive effect with short-term memory for in-frame/photomemristor compression, facilitating single-frame motion recognition and prediction, and power efficiency. A comparison of our photomemristor-based system with other reservoir computing systems and traditional vision sensors is given in Supplementary Table 1 and 2.'

In the revised manuscript, we have also included experiments on the memory-dependent accuracy of dynamic vision recognition (Figure 2f-h). In our system, changing the bias voltage across the Schottky barrier of the photomemristors tunes their inherent memory. The new results demonstrate that an increase of memory results in higher recognition accuracy. This memory-dependent accuracy effect, which resembles memory-dependent perceptions in the brain, could enable intelligent sensors with tunable attention.

Based on the above clarification and discussion, we hope that the reviewer could reconsider his/her assessment of our work.

Reviewer #4:

General comment

In this work by Tan et al., a retinomorphic in-sensor reservoir computing system is demonstrated based on a photomemristor array, which exhibits light-induced short-term memory with an accumulative effect. The ‘imprint’ of previous states serves as the basis of realizing various dynamic machine vision (DMV) tasks, such as single-frame dynamic vision recognition, single-frame motion recognition and prediction, and intelligent traffic simulation. This is an interesting work combining reservoir computing with DMV applications. However, I have the following questions for the preparation of a revised manuscript.

Our reply

We thank the reviewer for the positive assessment of our work and his/her suggestions to strengthen the manuscript. Below we address the comments on a point-by-point basis.

Reviewer comment 1

In the single-frame video recognition task, the authors compared the PMA with or without hidden states (previous memory), and the former achieved a much higher recognition accuracy. What I'm curious about is whether more remaining previous memory results in better accuracy. Can authors clarify this? Also, how are the normalized states in Fig. 2d obtained? Is it by normalization of the device current (I) or current change (ΔI)?

Our reply

To answer the reviewer's question about the remaining previous memory and its effect on the recognition accuracy, we repeated the experiment on video recognition shown in Fig. 2 but with a lower bias voltage of 0.8 V (less memory) and a higher bias voltage of 1.2 V (more memory) applied to the photomemristors. A revised Fig. 2 and new Supplementary Figure 11 (see below) show the new data. The results clearly indicate that the recognition accuracy increases with the remaining memory (Fig. 2f-h). In the main manuscript we added the following text on the new memory-dependent accuracy data (page 6-7):

‘In the brain, deep memory usually results in better perception. To evaluate the relation between memory and recognition accuracy in our RP-RC system, we tuned the inherent memory of the PMA by applying different bias voltages ($V_{\text{bias}} = 0.8 \text{ V}$, 1.0 V , and 1.2 V) across

the photomemristor Schottky junctions (Fig. 2f). Results for the same videos are shown in Fig. 2g and 2h and Supplementary Fig. 11. The data demonstrate that the recognition accuracy increases from 78% at 0.8 V to 100% at 1.2 V (Fig. 2h and Supplementary Fig. 11c) because of increased memory of previous frames (Fig. 2f and Supplementary Fig. 11b). This memory-dependent dynamic recognition, which resembles memory-dependent perception in the brain, could enable intelligent sensors with tunable attention.'

The normalized data in Fig. 2d was obtained by normalizing the photomemristor currents (I) that are recorded after the 5th frame. We added the following text to the caption of Fig. 2 to clarify this:

'The vectors are obtained by normalizing the photomemristor currents that are recorded after the 5th frame in (c).'

Revised Fig.2:

Figure 2. Memory-dependent dynamic vision recognition. a Videos playing ‘APPLE’, ‘LIME’, ‘OLIVE’, ‘DATE’, and ‘GRAPE’ letter-by-letter, all ending with ‘E’, are used as input to the retinomorph PMA. Only the photomemristor currents of the last frame (h_5) recorded after

playing the letter 'E' are used as vectors for recognition by the readout network. **b** Example of how the PMA memorizes the letter 'A'. After an illumination time of 100 ms, the letter 'A' fades in 2 s. **c** Output currents of the 25 photomemristors in the PMA recorded while playing the word 'A-P-P-L-E' letter-by-letter. The light pulses illuminate the PMA for 100 ms, the frame-to-frame rate is 2 Hz, and the bias voltage is 1 V. **d** Feature vectors of the last frame (letter 'E') for the five videos. The vectors are obtained by normalizing the photomemristor currents that are recorded after the 5th frame in (c). **e** Change in the photomemristor current of the last frame as a function of the number of previously received optical pulses. The error bars represent the standard deviation. **f** Photomemristor output current measured at different bias voltages ($V_{\text{bias}} = 0.8 \text{ V}$, 1.0 V , and 1.2 V) for optical pulses with a duration of 100 ms and a repetition rate of 2 Hz. The memory states (currents between optical pulses) increase with bias voltage. **g, h** Training and test accuracy for datasets recorded with different bias voltages in the same video classification task. The accuracy increases from 82% for training and 78% for testing at $V_{\text{bias}} = 0.8 \text{ V}$ to 100% for both training and testing at $V_{\text{bias}} = 1.2 \text{ V}$.

New Supplementary Fig. 11:

Supplementary Figure 11. Memory-dependent video recognition accuracy. **a** Schematic diagram of the RP-RC system with bias voltage dependent memory for recognition. The bias

voltages (V_{bias}) are 0.8 V, 1 V, and 1.2 V, corresponding to have low, medium, and high memory. **b** Photomemristor output currents (memory states) recorded at different bias voltages. The light pulses illuminate the PMA for 100 ms and the frame-to-frame rate is 2 Hz. The current states were read after the 5th pulse shown in Fig. 2f. The memory states increase with increasing bias voltage. **c** Confusion matrix of video recognition demonstrating an increase in test accuracy from 78% to 100% when the bias voltage is enhanced from 0.8 V (low memory) to 1.2 V (high memory).

Reviewer comment 2

The training and testing datasets in Fig. 2 are generated by adding noise to the experimental acquired current states rather than adding noise to the input pictures (e.g. <https://doi.org/10.1038/s41467-017-02337-y>). Could the authors discuss the effects of these two approaches?

Our reply

Adding noise to the input and output are both used in literature for training. The method of flipping pixels in the input pictures as used in <https://doi.org/10.1038/s41467-017-02337-y> is suitable for the classification of static binary images. In our experiments on the classification of videos (Fig. 2) and the recognition and prediction of motion (Fig. 3), we added Gaussian noise to the measured photomemristor currents. In the training of dynamic neural networks, the addition of Gaussian noise is more practical than the flipping of pixels in a sequence of images. Moreover, in the recognition and prediction of motion demonstrated in Fig. 3, the method of flipping pixels in the input pictures will not work because it involves spatiotemporal sequences of the same input.

We added the following clarification to the Methods section on page 14 of the manuscript: '1200 datasets (900 datasets for training and 300 datasets for testing) were generated by adding Gaussian noise to the recorded output currents for training. This method was chosen over the modification of input frames (i.e. via pixel flipping³⁰) because it is more practical for videos. The noise rates were 15% and 30%.'

REFERENCES

30. Du, C., Cai, F., Zidan, M. A., Ma, W., Lee, S. H. & Lu, W. D. Reservoir computing using dynamic memristors for temporal information processing. *Nat. Commun.* **8**, 2202 (2017).

Reviewer comment 3

The authors state that ‘Moreover, the uniform photomemristive response (Supplementary Fig. 3 and 4) guarantees high accuracy in various DMV tasks.’ It is nice to see the photomemristors with uniform responses, but for the single-frame dynamic vision recognition task, will the inhomogeneity of the device (D2D variability) necessarily reduce the accuracy?

Our reply

Device inhomogeneities do reduce the recognition accuracy. Training of the output, however, does limit the detrimental effects of inhomogeneities. This is demonstrated indirectly by the results shown in the revised Supplementary Fig. 10a-d. Here, we consider the added Gaussian noise to the photomemristor output current as a measure of the device inhomogeneity. Comparing the accuracies in Supplementary Fig. 10a, b (15% Gaussian noise) and Supplementary Fig. 10c, d (30% Gaussian noise), we can conclude that an increase in the noise rate from 15% to 30% lowers the accuracy by 6% (97.3% to 91.3%). On page 5 we added the following text:

‘Finally, we note that the photomemristive response of the PMA is highly uniform (Supplementary Fig. 4 and 5). While readout training could in part compensate for device-to-device variations in the output current, this hardware feature is relevant for complex DMV tasks.’

Revised Supplementary Fig. 10:

Supplementary Figure 10. Comparison of accuracies with different Gaussian noise (σ) and with/without hidden memory states. a and b Training and validation accuracy ($\sim 97.3\%$) of the readout network and confusion matrix for video recognition tasks with Gaussian noise factor $\sigma = 0.15$. The 2% misrecognition of ‘APPLE’ and ‘GRAPE’ is explained by both words containing the letters ‘A’, ‘P’, and ‘E’. **c, d** Training and validation accuracy ($\sim 91.3\%$) of the readout network and confusion matrix for the same video recognition tasks with Gaussian noise factor $\sigma = 0.30$. The $\sim 15\%$ misrecognition of ‘APPLE’ and ‘GRAPE’ is again explained by both words containing the letters ‘A’, ‘P’, and ‘E’. In **a-d**, the PMA worked as a retinomorph sensor, wherein dynamic memory states were used for the task of video recognition. **e** Training and validation accuracy ($\sim 36.2\%$) of the same readout network when the PMA works as conventional image sensor (using peak values of the photoresponse shown in Supplementary Fig. 1b) without hidden states (upper panels of Supplementary Fig. 9). The accuracy values are much lower than in **c**. **f** Corresponding confusion matrix of test video recognition with conventional photodetection, showing lower validation accuracy (22% – 48%) compared to **d**. The results in this figure demonstrate the importance of dynamic hidden states for dynamic vision perception.

Reviewer comment 4

The imprint factors in Fig. 3c are extracted from Fig. 3b. Could the authors give more detailed descriptions how the imprint factors are extracted? The imprint factor seems to be the bridge between experiments and simulations for the task in Fig. 4, and should be clearly elaborated.

Our reply

The imprint indicates the normalized memory of previous frames. The imprint factors in Fig. 3c are derived from Fig. 3b. The values are calculated by averaging the normalized memory of pixels #9,10,11,13,16,17,18,19,20,22,24,25 (the pixel indices are same as in Fig. 2b). The selected pixels are all illuminated by optical pulses representing the motion of the person. Fast motion results in a higher imprint factor, which is explained by a shorter decay time between frames. The differences in the imprint factor provide accurate speed classification (Fig. 3d). We added the following explanation on the extraction of the impact factors to the caption of Fig. 3:

'The imprint factors are calculated by averaging the normalized memory of pixels #9,10,11,13,16,17,18,19,20,22,24,25 (same indices as in Fig. 2b) in **b**. The selected pixels are all illuminated by optical pulses during object motion.'

REVIEWER COMMENTS

Reviewer #2 (Remarks to the Author):

I do not have any further comments. The authors addressed all my comments and I therefore believe that the manuscript can be a nice contribution to the journal.

Reviewer #4 (Remarks to the Author):

The authors have provided further results to address previous issues existing in the manuscript. I have no other questions and can recommend publication now.

Comments on manuscript entitled “Dynamic machine vision with retinomorph photomemristor-reservoir computing”

I thank the authors for addressing my comments and improving the manuscript, which now contains further demos on reservoir computing capability using their photomemristor array.

However, I still have major two comments:

1) This comment is related to an earlier one submitted in the previous review. Since in the rebuttal the authors agree that inference is achieved not by a single frame but multiple previous frames, phrases containing “single frame” in the manuscript should be removed or revised accordingly. Some locations where I still find such phrases are as follows:

- Abstract: “...single-frame motion recognition and...”; “...in-sensor single-frame motion processing...”
- Page 4: “...single-frame classification...”; “...MRP based on a single frame...”
- Page 5: “...demonstrate single-frame dynamic vision recognition...”
and several more.

In the video-playback of words demo, the authors chose words all ending with the same letter ‘E’. The demo shows that it is not the last frame containing ‘E’ but previous frames containing different letters that influence the inference. This is not single-frame classification. Similarly in the motion recognition and prediction demo, the autoencoder was trained using several frames. Besides, the demo was based on one dimension only. An object in a field of view could move in many different directions and many picture frames would be needed to train the model.

I believe what the authors actually mean is that processing of all frames occurs directly within the sensor or image plane. By virtue of its dynamic memory, the sensor can integrate information present in multiple frames to arrive at a cumulative results for inference. “Single frame”, on the other hand, carries a totally different meaning.

2) This comment arises from a new claim made by the authors in the revised manuscript. Page 4: “Here, we for the first-time exploit photomemristors for MRP based on a single frame with embedded memorized information from multiple previous frames.”

I beg to differ. Photomemristors have been exploited for motion recognition before. An example of a recent work can be found here: <https://doi.org/10.1002/aisy.202200196>. The authors did a very good job in augmenting the capability through more simulated demos. The underlying concepts and principles are, however, similar to earlier works. Appropriate revision should be made to the authors’ claim.

Overall, I think this manuscript makes a good contribution to the field. However, appropriate revisions are necessary to avoid misunderstanding and/or confusion.

Reviewer #2:

General comment

I do not have any further comments. The authors addressed all my comments and I therefore believe that the manuscript can be a nice contribution to the journal.

Our reply

We thank the reviewer for the positive assessment of our work and his/her recommendation of publishing our manuscript in Nature Communication.

Reviewer #3:

General comment

I thank the authors for addressing my comments and improving the manuscript, which now contains further demos on reservoir computing capability using their photomemristor array.

Our reply

We thank the reviewer for the evaluation of our work. Below we address the reviewer's comments on a point-by-point basis.

Reviewer comment 1

However, I still have major two comments:

1) This comment is related to an earlier one submitted in the previous review. Since in the rebuttal the authors agree that inference is achieved not by a single frame but multiple previous frames, phrases containing "single frame" in the manuscript should be removed or revised accordingly. Some locations where I still find such phrases are as follows:

- Abstract: "...single-frame motion recognition and..."; "...in-sensor single-frame motion processing..."
- Page 4: "...single-frame classification..."; "...MRP based on a single frame..."
- Page 5: "...demonstrate single-frame dynamic vision recognition..." and several more.

In the video-playback of words demo, the authors chose words all ending with the same letter 'E'. The demo shows that it is not the last frame containing 'E' but previous frames containing different letters that influence the inference. This is not single-frame classification. Similarly in the motion recognition and prediction demo, the autoencoder was trained using several frames. Besides, the demo was based on one dimension only. An object in a field of view could move in many different directions and many picture frames would be needed to train the

model. I believe what the authors actually mean is that processing of all frames occurs directly within the sensor or image plane. By virtue of its dynamic memory, the sensor can integrate information present in multiple frames to arrive at a cumulative results for inference. “Single frame”, on the other hand, carries a totally different meaning.

Our reply

We removed all phrases containing “single frame” from the manuscript.

Reviewer comment 2

2) This comment arises from a new claim made by the authors in the revised manuscript. Page 4: “Here, we for the first-time exploit photomemristors for MRP based on a single frame with embedded memorized information from multiple previous frames.” I beg to differ. Photomemristors have been exploited for motion recognition before. An example of a recent work can be found here: <https://doi.org/10.1002/aisy.202200196>. The authors did a very good job in augmenting the capability through more simulated demos. The underlying concepts and principles are, however, similar to earlier works. Appropriate revision should be made to the authors’ claim.

Our reply

We rephased the part highlighted by the reviewer to:

In recent years, photomemristors have been studied in neuromorphic vision and processing systems for image classification¹⁹⁻²³ and human action recognition²³. Here, we exploit photomemristors for MRP based on an informative frame with embedded memorized information from multiple previous frames and we demonstrate.

We included the recent work mentioned by the reviewer as Ref. 23.

General comment

Overall, I think this manuscript makes a good contribution to the field. However, appropriate revisions are necessary to avoid misunderstanding and/or confusion.

Our reply

We thank the reviewer for the assessment of our work and are confident that the revisions will avoid misunderstanding and/or confusion.

Reviewer #4:

General comment

The authors have provided further results to address previous issues existing in the manuscript. I have no other questions and can recommend publication now.

Our reply

We thank the reviewer for the positive assessment of our work and his/her recommendation of publishing our manuscript in Nature Communication.